# A Method for Measuring the Quality of Graphic Transfer to Materials with Variable Dimensions (Wood)

**DOI:** 10.3390/s22166030

**Published:** 2022-08-12

**Authors:** Renata Wagnerova, Martin Jurek, Jiri Czebe, Jan Gebauer

**Affiliations:** Department of Control Systems and Instrumentation, Faculty of Mechanical Engineering, VSB—Technical University of Ostrava, 70800 Ostrava, Czech Republic

**Keywords:** affine transformation, laser, quality control, wood, image processing

## Abstract

The transfer of graphics to a product’s surface is a widely known technology. Printing, engraving, and etching are used every day in production processes with countless types of materials. This paper deals with quality control for laser engraving on surfaces with variable dimensions via optical sensors. The engraving process, apart from colour changes, can induce volume and moisture changes, which lead to dimension changes in some materials. Natural materials and biomaterials are among the ones most affected. Combined with the porous and inhomogeneous structure of such a material, it can be difficult to measure the quality of graphic transfer, especially for shaded products. The quality control of laser-engraved photographs on thin layers of wood veneer was selected as a suitable problem to solve. A complex method for the quality measurement of the specified production was designed and tested. We used an affine transformation to determine the system behaviour and to determine the transfer function of material changes during the production process. Moreover, there is a possibility to compensate the image deformation of the engraved product.

## 1. Introduction

The transfer of graphics, marking, and design customisation of products have been increasing in value regarding customer needs for quite some time. It has found its meaning not only in production optimisation, tracking, or quality control, but also in product design, branding, and aesthetic value. Rising demand for high-end products made of renewable and biodegradable resources with eco-friendly [1] technologies has also been observed for some time now. As a result of this fact, it is becoming interesting to re-engage in natural materials, such as wood, even in areas where it has been replaced by plastics in the past or where it has not even been used. One of the leading eco-friendly technologies for marking and transferring graphics and photos to wood is laser engraving. Without the use of chemicals and ink, shaded graphics can be created using natural changes to the surface of the material [2,3] and can be precisely and easily formatted at the same time [4].

The inhomogeneity and dimensional instability of wood and similar materials are essential complications in achieving quality and repeatable results [5]. This changes mainly with moisture [6], which is due to its ability to absorb water [7,8]. During laser engraving, chemical processes occur that change the proportion of water in the material, and thus, dimensional changes occur during the production itself [9,10]. The amount of moisture and chemical composition [11,12] of wood influences the output of the laser-engraving process [13,14]. The wood veneers are dried during the production process to around 20%. Drying to 0% solves the problem with dimensional instability; however, this disturbs the material structure and lowers the lifetime of the product. As a result of this, it is a challenging task to measure and evaluate the quality of production (i) and to predict (ii) dimensional [15] and colour changes in the product during the ongoing production process [16,17]. Without this information, it is not easy to properly adjust production parameters [18,19].

This article is devoted to measuring dimensional changes which occur in materials with moisture absorption [20,21,22,23] and then evaluating the quality of the transferred graphics or markings. This makes it possible to better optimise production parameters and to achieve more precise dimensions (iii) and better production of the final product (iv) for specific wood species [24].

The study has the following contributions to the above-mentioned research gaps:A calibration system for dimensional changes during the production process via optical sensing. This system enables the comparison of production input and output even though the output is non-linearly deformed during production. See Section 2.2.A quality measurement system for the production of natural and inhomogeneous materials. This system enables the measurement and expressing of the quality for production with variable input materials (every wood veneer is different) and variable production input (every graphic is different). See Section 3.1.A study for a compensation system which can predict dimensional changes in the pre-production stage and can deform input graphics to compensate for material deformations and achieve desired results. See Section 2.2.A study on an expert system with a database which, with statistical data, can compare every production process (with variable materials and input graphics) and determine the best possible pre-production processing of the material, production parameters, and input graphics. See Section 3.3.

## 2. Equipment, Materials, and Experimental Procedures

Beech veneer samples were selected for testing the quality measurement method of graphic transfer to materials with variable dimensions. Beech wood is one of the most used types of wood in the wood industry [25]. It is widely used because of its availability and convenient properties [26]. It ranks as a hardwood with a typical density of 620–720 kg/m^3^. The samples were 1.4 mm thin and were sanded down with a P140 sanding disc grid. The surface of the wood sample is shown in Figure 1. The material surfaces were scanned with a high-resolution flatbed scanner equipped with a CIS optical sensor. The used sensor had an optical resolution of 4800 × 4800 dpi and 48-bit colour depth (8 bits per each colour).

### 2.1. Laser Machine

The testing machine was a common CNC laser-engraving machine with NEMA17 stepper motors and Pololu A4988 stepper motor drivers powered with a 12 V supply source. The motors were driven with a power of 1 A and were set to 1/16 microstepping mode. Both the motors and motor drivers were cooled down with passive heatsinks. The machine chassis was made of aluminium profiles and a Hiwin linear rail system. It also used a belt drive made from rubber and aramid fibres as internal reinforcement.

The laser-engraving module was based on the NUBM44-V2 blue (445 nm) laser diode. This laser emitter has exceptional beam power control features [27,28]. The maximal optical power output of this laser is around 7 W. The module was assembled out of a laser diode, three element glass lenses, and a copper body, which was used to transfer out the heat generated by the diode. The whole module was cooled down with a Peltier thermoelectric module with a rated power of 48.4 W and an additional air-cooled aluminium heat sink on the hot side of the Peltier module.

The laser emitter was placed 50 mm from the wood surface, and it was driven with a constant current device provided by the Polish company OPT-Lasers. A single channel laser driver with the designation LPLDD-5A-24V-PID-H has a powering range of 3–24 V with 5 A. These experiments were powered by a 12 V power supply. To dissipate power on the transistor, the driver was assembled with an anodised heat sink. The PID function was included and was used to maintain a constant temperature of the laser module via the Peltier module circuit. The driver provided a constant current power output of 4 V and 2.2 A with a soft start. PWM modulation was used to control the final output power and thus produced a colour shade within the engraving process. The driver output characteristic for a 10% PWM setting at 1 kHz, which corresponds to a pixel value of 25 in the picture with 8-bit colour depth (0–255), is shown in the graph in Figure 2.

An aspherical collimating lens made of three glass elements was selected to focus the 445 nm laser beam.

### 2.2. Affine Transformation

We used an affine transformation to find an optimal overlay of the source and laser-engraved images to improve our computation of the CIELAB standard deviation.

Generally, a geometric transformation maps points from one space to another. In our case, we used affine transformation, which preserves lines and does not commonly preserve parallelism [29]. Our goal was to estimate the transformation matrix W between the source and the scanned graphic (laser-engraved) images [1]. Both images used in our experiment can be seen in Figure 3a.

We selected the following graphic image for our experiment, which we amended with 14 landmarks in random positions, as seen in Figure 3b. The test image was laser engraved and finally scanned.

Let us divide our experiment into a few steps: landmark coordinate computing, affine transformation (AT) matrix computing, and AT parameter assessment with an analytic solution. We examined all the aforementioned steps in a MATLAB environment. 

The first was to manually select each landmark pair from the source and the scanned graphic image in a given order (see Figure 3b). MATLAB has a tool called Control Point Selection Procedure (CPSP), which is invoked via the command *cpselect*(). The CPSP tool enabled us to select the control points of two related images, or more precisely, landmarks. When the CPSP was opened, it allowed us to add, delete, and move landmarks interactively with the mouse. An illustration of how we used the CPSP tool is shown in Figure 4.

After the selection procedure, the tool saved all landmarks into two numeric vectors belonging to either moving or fixed images, i.e., the matrix *P_T_* of a moving (scanned) image and the matrix *P_R_* of a fixed (source) image. We used the MATLAB format for mathematical notation. Matrices *P_R_* and *P_T_* had the following form:(1)$$P_{R},P_{T}\in R^{n\times 2},P_{R}=\left[\begin{array}{cc} x_{1}^{R} & y_{1}^{R}\\ \vdots & \vdots \\ x_{n}^{R} & y_{n}^{R} \end{array}\right],P_{T}=\left[\begin{array}{cc} x_{1}^{T} & y_{1}^{T}\\ \vdots & \vdots \\ x_{n}^{T} & y_{n}^{T} \end{array}\right].$$
where pair (*x^R^*, *y^R^*) is the landmark coordinate of the source image, pair (*x^T^*, *y^T^*) is the landmark coordinate of the scanned (laser-engraved) image, and n is the total number of landmarks.

We planned to use affine transformation to perform scaling, shearing, rotation, and translation. Last was translation transformation, which was a nonlinear transformation in a 2D space to change it into a linear operation. We had to absorb the translation into an extra dimension. Using homogeneous coordinates, we extended our original 2D space into a 3D space, where each point’s 3rd coordinate was identity. Then, we could define affine transformation marked as *W* (4). The matrices *P_R_*, *P_T_*, and *W* had the following form:(2)$$P_{R}\in R^{n\times 3},P_{R}=\left[\begin{array}{ccc} x_{1}^{R} & y_{1}^{R} & 1\\ \vdots & \vdots & 1\\ x_{n}^{R} & y_{n}^{R} & 1 \end{array}\right],$$
(3)$$P_{T}\in R^{n\times 3},P_{T}=\left[\begin{array}{ccc} x_{1}^{T} & y_{1}^{T} & 1\\ \vdots & \vdots & 1\\ x_{n}^{T} & y_{n}^{T} & 1 \end{array}\right],$$
(4)$$W\in R^{3\times 3},W=\left[\begin{array}{ccc} w_{1} & w_{2} & t_{x}\\ w_{3} & w_{4} & t_{y}\\ 0 & 0 & 1 \end{array}\right].$$
where *W* is the transformation matrix, its elements *w*_1_, *w*_2_, *w*_3_, and *w*_4_ represent a linear transformation in a 2D space, *t_x_* is the translation on the x coordinate, and *t_y_* is the translation on the y coordinate.

We mapped the source image to the scanned image following the method approach presented in the literature [30]. Our target was to find the matrix *W*, such that
(5)$${\left\|P_{R}W\left(1,:\right)-P_{T}\left(:,1\right)\right\|}^{2}+{\left\|P_{R}W\left(2,:\right)-P_{T}\left(:,2\right)\right\|}^{2}\to \min.$$

Matrix coefficients *W* solved the problem
(6)$$P_{R}\cdot W^{T}=P_{T},$$
i.e., the mathematical problem
(7)$$\left[\begin{array}{ccc} x_{1}^{R} & y_{1}^{R} & 1\\ \vdots & \vdots & 1\\ x_{n}^{R} & y_{n}^{R} & 1 \end{array}\right]\cdot \left[\begin{array}{ccc} w_{1} & w_{3} & 0\\ w_{2} & w_{4} & 0\\ t_{x} & t_{y} & 1 \end{array}\right]=\left[\begin{array}{ccc} x_{1}^{T} & y_{1}^{T} & 1\\ \vdots & \vdots & 1\\ x_{n}^{T} & y_{n}^{T} & 1 \end{array}\right].$$

We obtained our approximation by the least-squares method
(8)$$W^{T}={\left(P_{R}{}^{T}\cdot P_{R}\right)}^{-1}\left(P_{R}{}^{T}\cdot P_{T}\right).$$

The resulting matrix *W* for the presented image couple was
(9)$$W=\left[\begin{array}{ccc} 2.3670 & 0.0264 & -0.9050\\ -0.0249 & 2.3728 & 247.0436\\ 0 & 0 & 1 \end{array}\right].$$

We could directly transform the source image into the scanned image space with the transformation matrix *W*. To perform the transformation in the opposite course, we had to determine the inverse matrix to *W*. The first step in confirming our results was to show landmarks of each image in a joint space. For example, we are using the source image space. The following Figure 5 displays the transformed landmarks.

A human factor caused a slight deviation of the landmarks during landmark selection. The average divergence in both axes was less than one pixel (px), particularly on the x-axis at 0.4 px and on the y-axis at 0.4 px.

Considering the slight error, our transformation of the scanned image had to fit very closely to the source image. The resulting inverse transformation matrix *W*^−1^ was:(10)$$W^{-1}=\left[\begin{array}{ccc} 0.4224 & -0.0047 & 1.5439\\ 0.0044 & 0.4214 & -104.1001\\ 0 & 0 & 1 \end{array}\right].$$

The application of the transformation to an image was achieved in MATLAB with the *imwarp*() function. The input arguments were an image in a supported format, a transformation matrix (10), and a reference size of an image (in the example, it is the size of the source image). The next step was to show our transformed scanned image with its source image in the joint space. MATLAB has a tool called *i**mshowpair*() to visualise two images with specific settings.

Figure 6 and Figure 7 show the resulting similarity between the source and the scanned image using different visualisation methods, such as blend (Figure 6a), diff (Figure 6b), and checkerboard (Figure 7). Each stated method has its benefits, e.g., we chose the blend method to show the overall overlay of both pictures. The diff method is excellent in gaining edges of laser-engraved landmarks. The method converts both images from RGB (colourmap) to greyscale and calculates the absolute difference. The output image from the method is also greyscale. We liked the checkerboard because it gives us an idea of how all graphic objects were laser engraved. It shows both images without an overlay, providing quick feedback about image deformation, the quality of laser engraving, or final similarity.

Figure 6a shows the overlay of both images, where the joint part is blended, and the outer parts of the source image, which was not the printout image, have a grey tone.

To fully display the similarity of both images, we displayed them on a checkerboard (see Figure 7a), where we selected a random area and showed its zoomed results, as seen in Figure 7b.

The resulting transform image was similar to its source. Figure 6b and Figure 7b show the zoomed areas that fully display an almost perfect fit.

Conclusion: Let us say that the transformation works precisely, but our outcome of improving the result is not visible yet. We want to decompose the resulting transformation *W* (9) into four transformations, such as scale (*S*), shear (*H*), rotation (*R*), and translation (*T*). Matrices *S*, *H*, *R*, and *T* have the following forms:(11)$$S=\left[\begin{array}{ccc} s_{x} & 0 & 0\\ 0 & s_{y} & 0\\ 0 & 0 & 1 \end{array}\right],$$
(12)$$H=\left[\begin{array}{ccc} 1 & h_{y} & 0\\ h_{x} & 1 & 0\\ 0 & 0 & 1 \end{array}\right],$$
(13)$$R=\left[\begin{array}{ccc} \cos\left(\varphi \right) & -\sin\left(\varphi \right) & 0\\ \sin\left(\varphi \right) & \cos\left(\varphi \right) & 0\\ 0 & 0 & 1 \end{array}\right],$$
(14)$$T=\left[\begin{array}{ccc} 1 & 0 & t_{x}\\ 0 & 1 & t_{y}\\ 0 & 0 & 1 \end{array}\right].$$
where *t_x_* and *t_y_* represent a translation of a point in pixels on the axes x and y, *φ* is an angle rotation around the origin in radians, *h_x_* is a horizontal shear factor, *h_y_* is a vertical shear factor, and *s_x_* and *s_y_* scale the x and y coordinates of a point.

Let us assume that our founded transformation matrix *W* (9) consists of the transformations mentioned above (matrices *T*, *R*, *S*, *H*), i.e.,
(15)W=T·R·S·H.

We can directly express the translation (*t_x_* and *t_y_*) from the transformation matrix *W* (4), (9). We can focus on the 2D space transformations, i.e., on parameters *w*_1_, *w*_2_, *w*_3_, and *w*_4_. The parameters are:(16)$$w_{1}{=s}_{x}\cdot \cos\left(\varphi \right)-h_{x}\cdot s_{y}\cdot \sin\left(\varphi \right),\;w_{2}{=h}_{y}\cdot s_{x}\cdot \cos\left(\varphi \right)-s_{y}\cdot \sin\left(\varphi \right),\;w_{3}{=h}_{x}\cdot s_{y}\cdot \cos\left(\varphi \right){+s}_{x}\cdot \sin\left(\varphi \right),\;w_{4}{=s}_{y}\cdot \cos\left(\varphi \right){+h}_{y}\cdot s_{x}\cdot \sin\left(\varphi \right).$$

The previous comparison tests of laser-engraved images exhibited slight shearing on the x-axis. Let us assume that parameter *h_x_* is negligible, i.e., equal to zero. Another solution constraint relates to the rotation angle *φ*, expected in the range (−π, +π). The last solution constraint ensures a positive value of scale parameters *s_x_* and *s_y_*. All solution constraints are:(17)$$h_{x}=0,\;\varphi \in \left(-\pi ,+\pi \right),\;s_{x}>0,\;s_{y}>0.$$

We use the second-order Taylor polynomial approximation (see Figure 8) instead of the harmonic function sine and cosine. The Taylor polynomials are
(18)$$\sin\left(\varphi \right)=\overset{k}{\underset{i=0}{\sum }}\frac{{\left(-1\right)}^{i}}{\left(2\cdot i+1\right)!}\cdot \varphi^{2\cdot i+1},$$
(19)$$\cos\left(\varphi \right)=\overset{k}{\underset{i=0}{\sum }}\frac{{\left(-1\right)}^{i}}{\left(2\cdot i\right)!}\cdot \varphi^{2\cdot i},$$
where *k* represents an order of Taylor polynomial approximation.

The final form of our solution (16) after the substitution of parameter *h_x_* and the second-order Taylor polynomial is:(20)$$w_{1}=s_{x}\cdot \left(\frac{\varphi^{4}}{24}-\frac{\varphi^{2}}{2}+1\right),\;w_{2}=h_{y}\cdot s_{x}\cdot \left(\frac{\varphi^{4}}{24}-\frac{\varphi^{2}}{2}+1\right)-s_{y}\cdot \left(\frac{\varphi^{5}}{120}-\frac{\varphi^{3}}{6}+\varphi \right)\;w_{3}=s_{x}\cdot \left(\frac{\varphi^{5}}{120}-\frac{\varphi^{3}}{6}+\varphi \right),\;w_{4}=s_{y}\cdot \left(\frac{\varphi^{4}}{24}-\frac{\varphi^{2}}{2}+1\right){+h}_{y}\cdot s_{x}\cdot \left(\frac{\varphi^{5}}{120}-\frac{\varphi^{3}}{6}+\varphi \right)$$

Now we can find our analytical solution (20) using the specified constraints (17). We can see the resulting decomposition of our transformation matrix *W* (9) into scale, rotation, shear, and translation transformation. The resulting parameters are:(21)$$s_{x}=2.3672,s_{y}=2.3729,\;h_{x}=0,h_{y}=0.0006,\varphi =-0.0105\;\mathrm{rad},\;t_{x}=-0.9050\;\mathrm{px},t_{y}=247.0436\;\mathrm{px}.$$

### 2.3. Quality Measurement Method

Production quality is essential for every type of product. To control it, an effective and quick quality measurement method for the transfer of images onto wood surfaces was designed and tested. The method consists of three distinguished steps. 

The first step is wood surface sampling. A high-resolution scanner is used to scan the resulting graphic on the wood surface. The scan is equipped with a shade balance checker, which consists of true white (255), black (0), and grey (125) colours and is placed next to the wood sample to calibrate the scanner for every sample [31]. This allows for quality measurement and wood species identification [32].

The second part deals with the dimensional instability of porous materials. Due to the heat produced by laser-engraving technology, dimensional change is expected for thin samples. Affine transformations introduced in the previous subchapter are used to precisely match the pixels of the source graphic with the pixels of the scanned graphic.

The third step is to measure and calculate the results of the produced graphic transfer. Calculations are made in the MATLAB environment. The dependence of the colour shades of the pixels between the source graphic and the produced scan is evaluated. To measure the precision of the provided results, the standard deviation of pixel matches is calculated. 

The quality of image transfer is described by several key factors. They are the linearity of shade characteristics (LAB colour space), the darkest shade achieved, the lightest shade achieved, and the standard deviation of shade dependencies.

## 3. Results and discussion

### 3.1. Quality Measurement Method

To better identify quality differences among multiple products with different graphics, a comparative method was designed.

This method was specially designed for engraving photographs and graphics into inhomogeneous materials. The unambiguous determination of quality is challenging because each product is an original piece. Both the input graphics and the structure of the wood (drawing and shade) enter the quality process. It is not easy to push the quality of production forward when the input parameters change. To assess the quality of the production process independently of the input parameters, general and statistical approaches were used for uniform evaluation. The main input to the evaluation is the input graphics and the scan of the resulting image, from which pairs of pixels are created, which are related to each other. These pairs are large enough to be able to average the relationship between all the individual shades, which are in the input graphics 255. The number of pixel pairs used for evaluation is for this sample of veneer with dimensions of 300 mm × 200 mm with a total of 6,000,000.

Quality is evaluated using five basic parameters that are selected; therefore, they can be observed for input graphics variables. These are the ratio of linearity, the lightest shade, the darkest shade, the overall depth of achieved shades, and the standard deviation of the calculations. The final products are sought with the greatest possible depth of shades and the most linear production process possible; therefore, the product corresponds to its specifications as much as possible. Values of the lightest and darkest shades are also taken into account, since the contrast is very important for achieving sufficient details of input graphics [33]. The standard deviation must be as low as possible in the evaluation, as higher numbers lead to uncertain results, even for averages performed from every pixel value.

The evaluation of the quality of shades is processed in the MATLAB program and is divided into four basic parts. The first part is the processing of input graphics and scans. Then, the calculation of L*a*b* characteristics takes place with the subsequent calculation of standard deviation, and everything is finalised by quality calculations with subsequent graphing.

Before the input photography and the resulting scan are loaded into MATLAB, they are restored and cropped using the affine transformation described in Section 2.2. Graphics with identical dimensions are uploaded to the evaluation process, which are accurately projected pixel by pixel.

Calculations of hue characteristics take place by gradual iteration, pixel by pixel. All pixel values of the input graphic (1–255) are assigned to corresponding pixel values of the scanned product. These values are then averaged, and dependency waveforms are created for all CIELAB colour space axes. The results of these measurements, which represent the common characteristic of wood burning by increasing the power of the laser emitter [2], are shown in the following graphs.

The CIELAB a* characteristic tends to move slightly from the green spectrum to the red spectrum. This is because the burned colour of wood varies in shades of brown to black. [34] It is visible in Figure 9.

The CIELAB b* characteristic shifts slightly from blue to yellow, as can be seen in Figure 10. Obviously, changes in the colour cast of shades change in both CIELAB a* and b* characteristics similarly.

The CIELAB L characteristic, the general lightness of the resulting shades, ranges non-linearly from light to dark depending on the input graphics, as shown in Figure 11. Since this characteristic is purely linear for the input graphics, the higher quality of the transmission of graphics to the wood surface is due to the effort of maintaining this linearity.

Thus, this graph is additionally subjected to linearity measurements. This is performed by interlacing the first-order directive and then by evaluating the areas that are either missing or remaining in this graph. These areas are then divided by the resulting area under the curve, and their share is expressed as a percentage, where 0% is no match and 100% is complete linearity. The evaluation of such linearity is shown in Figure 12.

The standard deviation is calculated for all individual shades to show how much the scanned values differ from each other. This is due to possible inaccuracies in the overlap of individual graphics, as well as the natural drawing of wood. The influence of the inhomogeneity of the wooden base is examined in more detail in Section 3.2.

To monitor the quality of engraved wood photographs, a total of 20 individual samples were produced, which were averaged, and the individual results were compared against the overall average to show which samples achieved what results and in what indicator. Of these, the first 10 samples were ground and sufficiently dried. A further 10 samples were additionally treated with sodium bicarbonate in addition before drying.

### 3.2. Standard Deviation

The standard deviation in these calculations should point to inaccuracies caused by the natural structure of the wood, to inaccuracies in the shift and tilt of the scan and input graphics, and to possible inhomogeneities in production. The entire quality measurement is based on averaging individual pairs of values. Obviously, for a natural process such as wood burning, there is always some thermal influence on the surroundings of the engraved point that can bring inaccuracy to the measurement. Therefore, it is necessary to visualise not only the average value of the standard deviation, but also its course. Its appearance regarding the technology solution is depicted in Figure 13.

The standard deviation values in this instance range from 0 (which is never realistically reached) to around 73 (which represents absolutely no correlation between the source and scanned image; in this instance, all individual pixel values of the source image have their corresponding pixels with every possible value in the scanned image with a symmetrical distribution). Generally, the standard deviation for this task is around 5–8 and should not exceed a value of 10. Higher values indicate a poor match quality of source and scanned graphics. The final number consists of the average of standard deviations for each pixel value, and this indicator is part of the quality evaluation process. As the value becomes smaller, the course of both production and the resulting evaluation becomes better.

It should be noted that the structure of the wood has an influence on the resulting measurement; however, this influence can be overlooked with sufficiently large averaging, and so it is possible to neglect this influence for the final comparison of individual samples with each other.

### 3.3. Quality Improvement by Production Parameter Optimisation

As a result of this method for evaluating the quality of graphics transferred to wood using laser engraving, it is possible to compare individual products with each other, even though their appearance and processing are completely different. As a result of this, it is possible to deploy a continuous system for quality management, which can allow a gradual increase in quality during ongoing production. If additional systems are used to monitor the influence of input parameters and the state of the wood, then the quality of the technology solution increases with each subsequent product. Elements for which the human eye in production cannot suffice are gradually uncovered, and it is possible to describe this technology and achieve maximum results for each photo. As a result of the fact that it is a technology whose result has several degrees of freedom and because the quality in the future depends on the evaluation of thousands of samples, the deployment of gradual methodologies is the only way to ensure a quality result and, above all, a repeatable one. As already mentioned in Section 3.1, the individual parameters of the resulting quality are compared against the obtained average of all calculated samples. The result is defined in Equation (22).
(22)$$\mathrm{Quality}=\left(\left(\frac{S_{L}}{A_{L}}-1\right)+\left(1-\frac{S_{DS}}{A_{DS}}\right)+\left(\frac{S_{LS}}{A_{LS}}-1\right)+\left(\frac{S_{E}}{A_{E}}-1\right)\right)\cdot \left(\frac{A_{S}}{S_{S}}\right)$$

Equation variables are marked such that *S* represents the value of the measured sample and *A* represents the average value of previous datasets. Brackets are distributed with indexes *L* (linearity), *DS* (the darkest shade), *LS* (the lightest shade), *E* (shade depth), and *S* (standard deviation).

The database of the resulting measurements are constantly replenished and recalculated according to new measurements. As a result of this, it is possible to gradually compare thousands of samples with each other at once, and it is clear not only which samples are better than others, but also exactly why. A visualisation of this evaluation is shown in Figure 14.

Samples 210001–210007 were only dried prior to production, and samples 210008–210015 were treated with sodium bicarbonate prior to drying. From the column of overall rating, it is clear at first glance that the samples that were treated with sodium bicarbonate before production and drying achieved better results in the proposed quality measurement. This was due to its ability to achieve darker shades, which were caused by the presence of a greater concentration of black carbon. As a result of this, they also had a greater depth of achieved shades, and the created shades were more homogeneous, which was defined by a lower value of standard deviation.

It is important to mention that the methodology for evaluating the resulting quality was based on two logical assumptions. Firstly, the lightest and darkest finishes—and therefore the resulting depth—have a greater impact on the final quality than the linearity of the shades. This is especially important for vector graphics, which, unlike raster graphics, often do not use the full range of shades. The system is designed to be robust and to be able to work for both black and white images and greyscale images. Secondly, the system counts on many samples in the future, and samples that achieve high standard deviations are forcibly suppressed, since they contain high uncertainty in their measurements. Moreover, when using additional evaluation systems, it is better to suppress such samples, so results are not based on uncertain measurements.

### 3.4. Quality Improvement by Implementing a Calibration Process Based on Affine Transformation

The significant and usable parameter from decomposition is the scale (*s_x_*, *s_y_*). We can use the difference on both axes to transform the original image, i.e., perform controlled deformation. Our goal is to reduce the final image ratio deviation between an original and laser-engraved image. The process validation is as follows in Figure 15.

Let us define the scale matrix *S_CD_* (23), which performs controlled deformation (transformation) of the source image landmarks. *S_CD_* has the form:(23)$$S_{CD}=\left[\begin{array}{ccc} 1 & 0 & 0\\ 0 & \frac{s_{x}}{s_{y}} & 0\\ 0 & 0 & 1 \end{array}\right]=\left[\begin{array}{ccc} 1 & 0 & 0\\ 0 & 0.9976 & 0\\ 0 & 0 & 1 \end{array}\right],$$

The next step is to apply *S_CD_* transformation to source image landmarks *P_R_* (2) to obtain a new set of deformed landmarks *P_RCD_*, as seen in Equation (25). The *P_RCD_* matrix has the form:(24)$$P_{RCD}\in R^{n\times 3},P_{RCD}=\left[\begin{array}{ccc} x_{1}^{RCD} & y_{1}^{RCD} & 1\\ \vdots & \vdots & 1\\ x_{n}^{RCD} & y_{n}^{RCD} & 1 \end{array}\right],$$
(25)$$P_{RCD}{=P}_{R}\cdot S_{CD}^{T}.$$

Let us assume that transformation matrix *W* (9) is the transfer function of the laser-engraving machine. We must repeat our experiment with a different reference (source) image represented with *P_RCD_* landmarks (24). We can simulate our experiment by applying the device’s transfer function to *P_RCD_* landmarks to see how the laser-engraving machine would print it (27). *P_TCD_* landmarks represent the resulting simulated image. The *P_TCD_* landmark matrix has the form:(26)$$P_{TCD}\in R^{n\times 3},P_{TCD}=\left[\begin{array}{ccc} x_{1}^{TCD} & y_{1}^{TCD} & 1\\ \vdots & \vdots & 1\\ x_{n}^{TCD} & y_{n}^{TCD} & 1 \end{array}\right],$$
(27)$$P_{TCD}=P_{RCD}\cdot W^{T}.$$

The last step is to verify the data. We must find a new affine transformation between source image landmarks *P_R_* (2) and the simulated print represented by *P_TCD_* landmarks (26). We expect to reach almost identical values of *w_1_* and *w_4_* parameters of our recent (controlled deformation) transformation matrix *W_CD_* with the following form:(28)$$W_{CD}\in R^{3\times 3},W_{CD}=\left[\begin{array}{ccc} w_{1} & w_{2} & t_{x}\\ w_{3} & w_{4} & t_{y}\\ 0 & 0 & 1 \end{array}\right].$$

Our target is to find the matrix *W_CD_*, such that
(29)$${\|P_{R}W_{CD}\left(1,:\right)-P_{TCD}\left(:,1\right)\|}^{2}+{\left\|P_{R}W_{CD}\left(2,:\right)-P_{TCD}\left(:,2\right)\right\|}^{2}\to \min.$$

Matrix coefficients *W_CD_* solve the problem
(30)$$P_{R}\cdot W_{CD}^{T}=P_{TCD}.$$

The resulting matrix *W_CD_* has the following form
(31)$$W_{CD}=\left[\begin{array}{ccc} 2.3670 & 0.0263 & -0.9050\\ -0.0249 & 2.3670 & 247.0436\\ 0 & 0 & 1 \end{array}\right].$$

We can directly compare the *W_CD_* matrix (31) with the original *W* matrix (9). The *W_CD_* matrix demonstrates that our approach is correct. Parameters *w*_1_ and *w*_4_ are almost identical. The deviation of those parameters was found to be 1.5 × 10^−5^. Applying our ascertained controlled deformation matrix *S_CD_* to the source image, we expect to obtain better laser-engraving results from our machine.

We presented the first draft of how we plan to use an affine transformation to calibrate the laser-engraving machine. In the future, we will schedule a large test set of engraved images to detect all parameters of the transformation matrix with presented decomposition. From the current results, we can only work with scale parameters when we are planning to adjust an original image to ensure a lower deviation of the laser-engraved image to final image ratio. We expect to perform machine calibration for each printing material or re-calibration for large image bundles in the future.

## 4. Conclusions

Laser-engraving technology is being used more and more in today’s world. The ability to look beyond transmission mediums (paint, ink) is not only more efficient, but also more environmentally friendly. It is a complicated technology, and if it is used for inhomogeneous materials, its application becomes even more complicated. In the case of natural materials such as wood or leather, the situation is yet even more complicated, as each piece of the base material is different. It is not only about structure, but also about physical properties such as age (chemical composition), humidity, and others. A method that would help measure quality and evaluate the settings of this technology and pre-production stages has been designed and tested in production. 

A calibration system for dimensional changes during the production process via optical sensing was designed and tested. As a result of this system, it is possible to compare the production’s input and output, even though the output was non-linearly deformed during production. An affine transformation was used to transform the original input graphic to precisely fit the scanned graphic (deformed wood veneer). This task can also be performed manually with some degree of experience; however, it takes around 10–15 min. With a minimum of three landmarks (small crosses in the corners of the graphic) and affine transformation calculations, the task can be performed automatically, which is essential for any kind of expert system where thousands of samples are needed for proper results.

A quality measurement system for the production of natural and inhomogeneous materials was designed and tested. This system enabled the quality of the production of different graphics on variable materials (every piece of wood veneer and input graphic is different) to be quantified. The inhomogeneity of wood surface structures and dimensional instability were overcome with standard deviation calculations and affine transformations (gap 1). A quick method, which considers every single pixel of the input and scanned graphic, was written in MATLAB language. This provides millions of calculations which, with proper averaging, ensures relevant data for quality control.

A study for a compensation system which can predict the dimensional changes in the pre-production stage and which can deform input graphics to compensate for material deformations and achieve desired results was presented. This possibility was discovered during ongoing research. After the first tests, the possibility to predict deformations based on humidity measurements and wood structure scans (count and distribution of tree growth rings) looked realistic. However, solving this task requires another expert system with hundreds of production cases. It should be noted that this system can provide greater precision, since it not only measures the deformation and adjusts to it, but it can also compensate for it before production. Work on this complex issue requires additional time. 

A study on an expert system with a database which can, based on production data, determine the optimal pre-production processing, production parameters, and input graphics adjustments was designed and discussed. A sufficient way to compare and highlight productions with the worst and best results was designed. After the first testing, it was determined that this approach can work and can bring results which are needed for quality optimisation. However, the pre-production and production setting data need to be included in the final evaluation in the database. With this, it can be clear not only how individual productions turn out (current state), but also why they turn out this way (future testing). This is only possible with thousands of measured and described production cases. That is why the current design uses an automated calibration system for precise matching of the input and scanned graphics and automated quality measurement system. With only the input graphics and the scan of the final product acquired with the optical CIS sensor, all primary data are automatically created, calibrated, calculated, and uploaded to a cloud database, which immediately recalculates itself accordingly. 

During this study, four main scientific gaps/problems were presented. A calibration system for dimensional changes and a quality measurement system were designed, tested, and implemented into the production of laser-engraved photographs. These systems can be implemented into other productions where dimensional changes or variable laser-engraving production occur. A study on an expert system with a database was designed and tested. It proved to be working as a concept; however, additional data are required to utilise its full potential. A study for a compensation system was discovered and discussed. It is yet to be tested, however, after a preliminary study indicated that, with the power of affine transformation, it can be solved. It is a similar task to the expert system, which also deals with thousands of samples via an automated system. It is essential to autonomously evaluate samples among themselves [24], not only based on the final quality, but also on the basis of pre-production stages, humidity, and the wood structure.

## Figures and Tables

**Figure 1 sensors-22-06030-f001:**
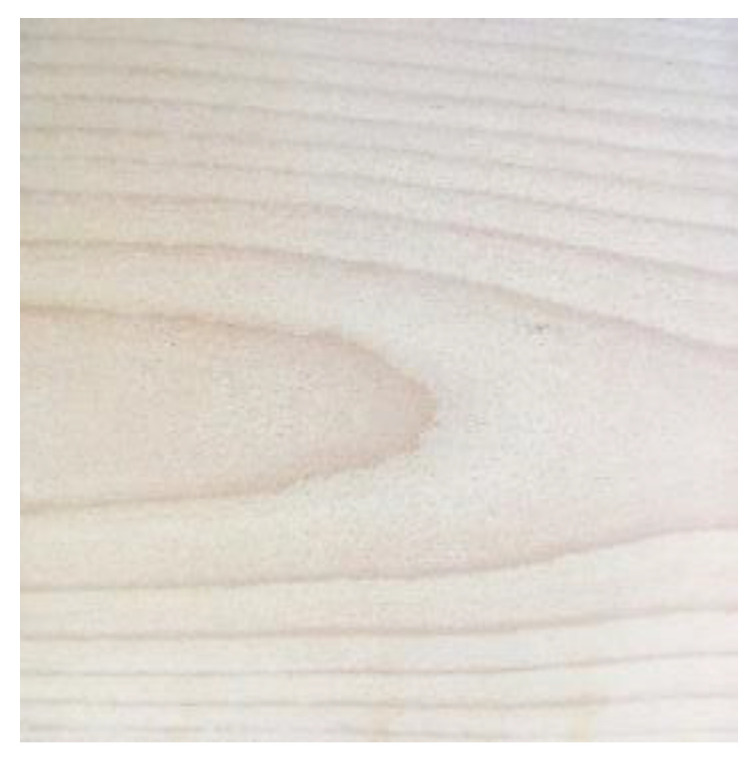
Beech wood surface.

**Figure 2 sensors-22-06030-f002:**
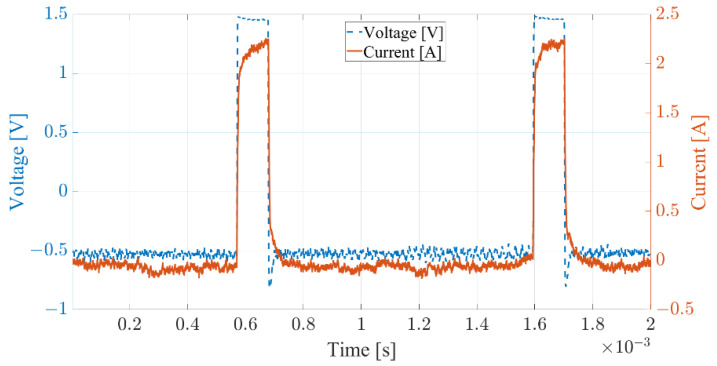
Power output of laser driver.

**Figure 3 sensors-22-06030-f003:**
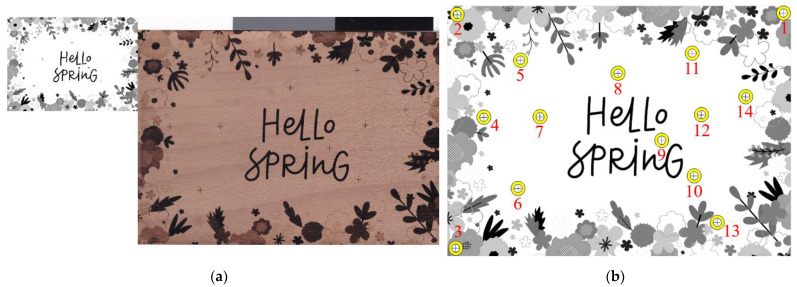
Source and scanned (laser-engraved) image: (**a**) From the left side source and scanned image preview; (**b**) Source image’s landmarks.

**Figure 4 sensors-22-06030-f004:**
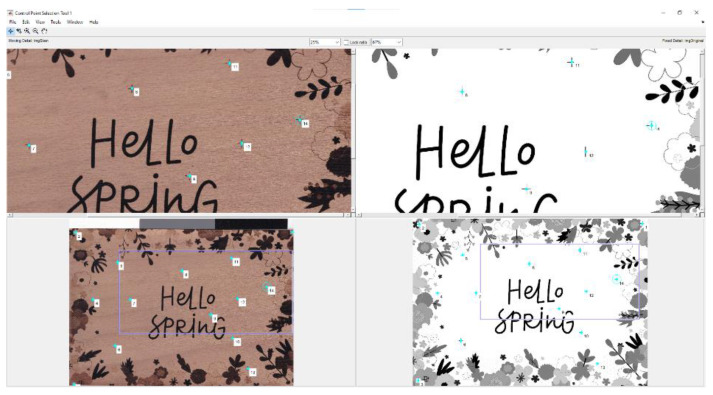
GUI of CPSP tool.

**Figure 5 sensors-22-06030-f005:**
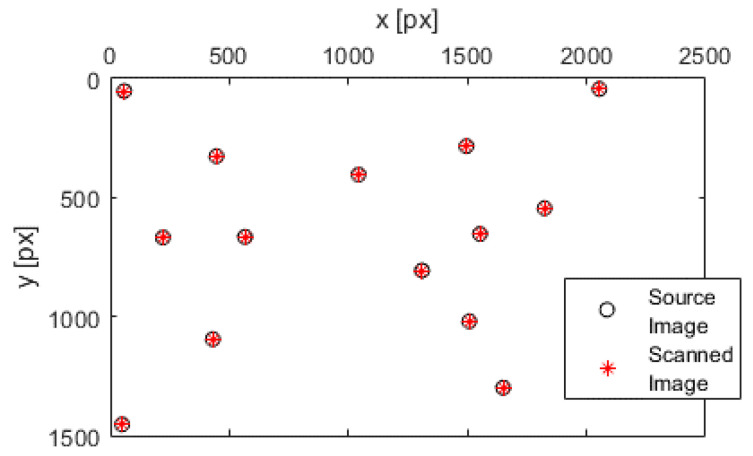
Source and scanned image landmarks (joint space).

**Figure 6 sensors-22-06030-f006:**
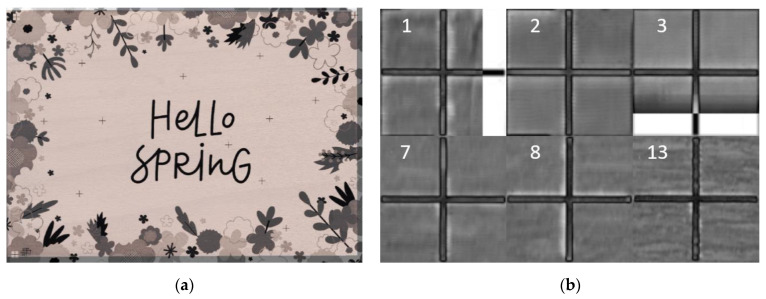
Scanned and source image comparison method: (**a**) Blend method; (**b**) Diff method, zoomed selected landmarks (position of landmarks see Figure 3b).

**Figure 7 sensors-22-06030-f007:**
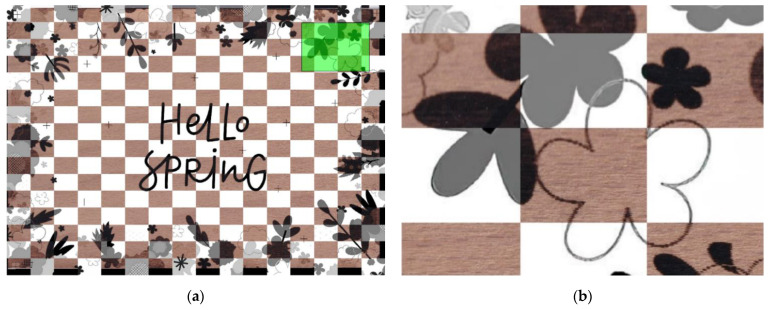
Scanned and source image comparison method: (**a**) Checkerboard method; (**b**) Zoomed area.

**Figure 8 sensors-22-06030-f008:**
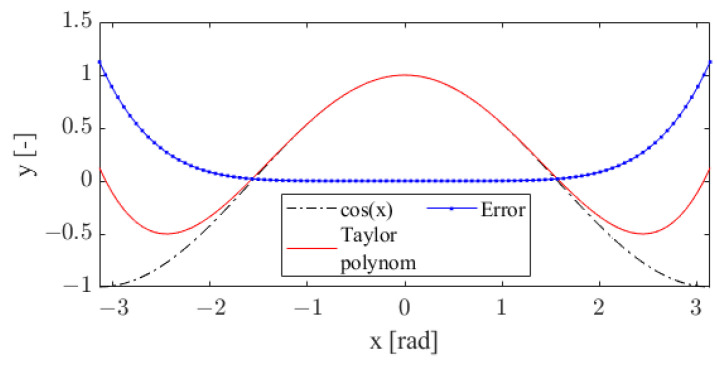
Second-order Taylor approximation of a cosine function.

**Figure 9 sensors-22-06030-f009:**
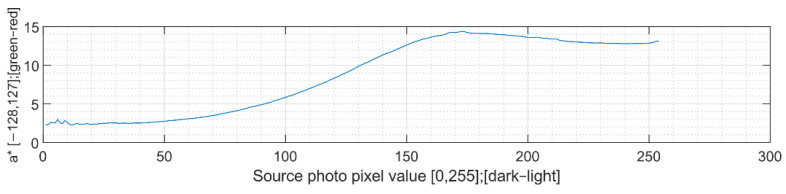
CIELAB a* characteristic of wood-engraved picture.

**Figure 10 sensors-22-06030-f010:**
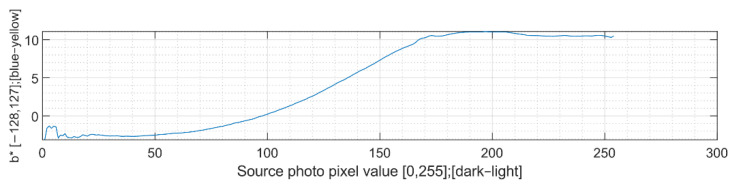
CIELAB b* characteristic of wood-engraved picture.

**Figure 11 sensors-22-06030-f011:**
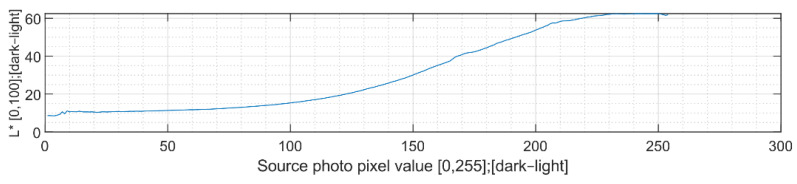
CIELAB L characteristic of wood-engraved picture.

**Figure 12 sensors-22-06030-f012:**
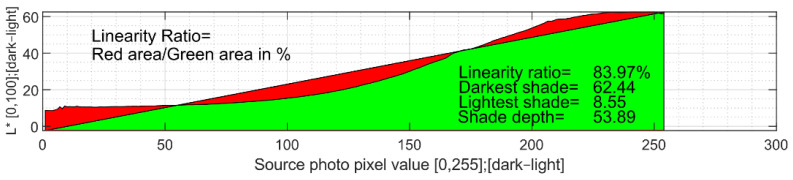
Linearity of shade dependence evaluation.

**Figure 13 sensors-22-06030-f013:**
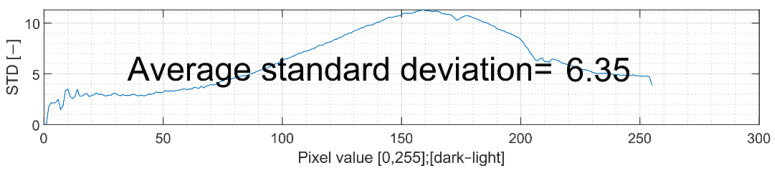
Standard deviation characteristic of matched graphic calculations.

**Figure 14 sensors-22-06030-f014:**
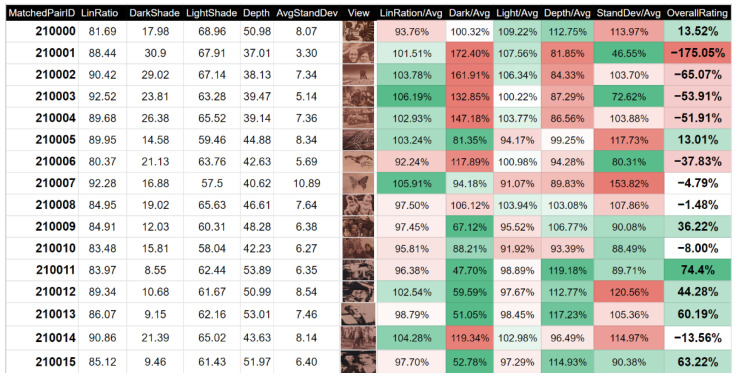
Quality calculations for scanned engraved graphics.

**Figure 15 sensors-22-06030-f015:**
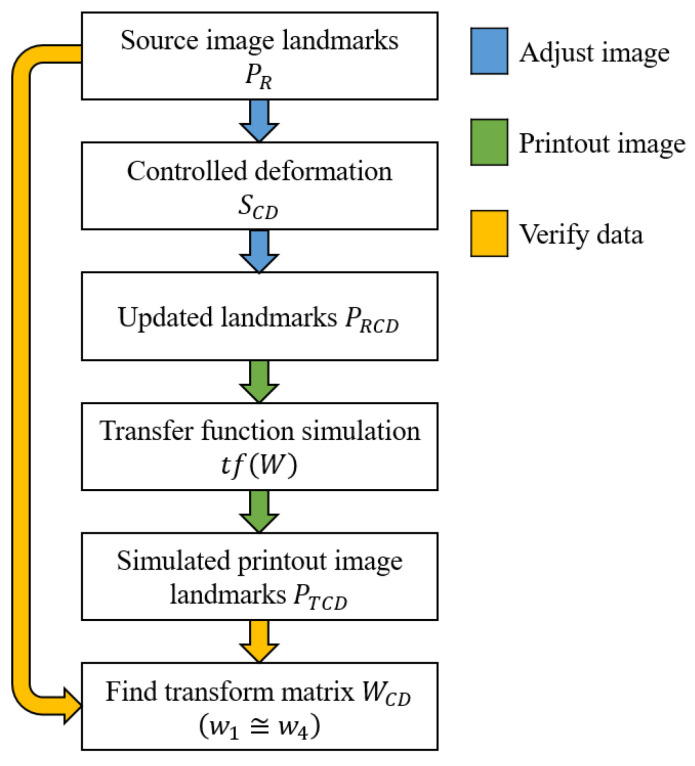
Calibration process validation.

## Data Availability

Not applicable.

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
