# Peer review of "A Method for Measuring the Quality of Graphic Transfer to Materials with Variable Dimensions (Wood)"

_sensors, 2022, doi:10.3390/s22166030_

Round 1

Reviewer 1 Report

The paper “Method for measuring the quality of graphic transfer to materials with variable dimensions (wood)” deals with quality control for laser engraving on surfaces with variable dimensions via optical sensors. And it can compensate the image deformation of the engraved product. There are some doubts.

1. In the introduction and conclusions of this paper, the influence of water absorption and humidity of wood on Yield and quality is mentioned, but the paper does not start to say how to solve this problem.

2. In section 2.2, Figure 4 shows how to use CPSP tool, but it does not explain in detail what work is done with CPSP tool and what functions are realized.

3. In section 2.2, the effect of laser temperature on the structure of the wood and thus on the final experimental results is not reflected in the next text.

4. Can “the diff method” mentioned in Figure 6 (b) be expanded to introduce what functions it can achieve?

5. The three steps of the quality measurement method in section 2.3 should be carried out specifically to explain how to implement the steps.

6. The structure of wood will affect the measurement result, but why is this effect linear?

7. In line 305, it is written that the structural effects of wood are linear, why the effects are linear. And why are heterogeneous materials more complicated?

8. In section 3.2, the value of standard deviation is 6.3569. But in this paper, there was no qualified value of standard deviation. It is not clear whether this value fits the bill.

Author Response

Dear Reviewer,

Thank you for your review and suggestions; we have changed our text accordingly and have highlighted these changes in our revised paper.

Reviewer 2 Report

Report on the manuscript

Title:  Method for measuring the quality of graphic transfer to materials with variable dimensions (wood)

Renata Wagnerova, Martin Jurek, Jiri Czebe and Jan Gebauer

ID sensors-1816499

I think the readers of this journal will appreciate the results of this manuscript.  This paper deals with quality control for laser engraving on surfaces with variable dimensions via optical sensors. The engraving process, apart from color changes, induces volume and moisture changes which lead to dimension changes in some materials. A complex method for quality measurement of the specified production was designed and tested. Generally speaking, the manuscript is well written, the material is judiciously divided and organized and correct from scientific point of view. Some changes are, however, necessary. For these reasons I can recommend the acceptance of this paper after some corrections presented in the attached file

Author Response

Thank you for your valuable comments. In terms of the revisions made, we have taken your constructive suggestions and have made modifications to the paper. We have also highlighted these changes in the paper. 

Round 2

Reviewer 1 Report

The author has answered all the questions, so it is suggested to be accepted.

Reviewer 2 Report

No comments